# An atlas of human proximal epididymis reveals cell-specific functions and distinct roles for CFTR

Shih-Hsing Leir, Shiyi Yin, Jenny L Kerschner, Wilmel Cosme, Ann Harris

**Spermatozoa released from the testis are unable to fertilize an egg without a coordinated process of maturation in the lumen of the epididymis. Relatively little is known about the molecular events that integrate this critical progression along the male genital ducts in man. Here, we use single cell RNA-sequencing to construct an atlas of the human proximal epididymis. We find that the CFTR, which is pivotal in normal epididymis fluid transport, is most abundant in surface epithelial cells in the efferent ducts and in rare clear cells in the caput epididymis, suggesting region-specific functional properties. We reveal transcriptional signatures for multiple cell clusters, which identify the individual roles of principal, apical, narrow, basal, clear, halo, and stromal cells in the epididymis. A marked cell type–specific distribution of function is seen along the duct with local specialization of individual cell types integrating processes of sperm maturation.**

## Introduction

The human epididymis has a pivotal role in male fertility. Immature sperm leaving the testis are exposed to a series of key environmental cues in the lumen of the duct that ensure their full maturation. These cues are provided in large part by cells in the epithelium of the epididymis, which secrete a complex mixture of ions, glycoproteins, peptides, and microRNAs (Belleannee et al, 2012a) that coordinate sperm maturation along the length of genital ducts. Most insights into the functional specialization of the epididymis epithelium arise from studies on rodents (primarily mouse and rat) and larger mammals such as the pig (Jervis & Robaire, 2001; Robaire & Hinton, 2002; Dacheux et al, 2005; Dacheux et al, 2009; Breton et al, 2016). However, it is apparent there are substantial differences between species, both in structure and detailed functions. Knowledge of the human male genital ducts is less well advanced because of the difficulty of obtaining live tissues for research and the impossibility of performing in functional studies in vivo. Anatomical observations show that unlike in rodents, where the different functional zones of the epididymis, the

initial segment, the caput (head), corpus (body), and cauda (tail) are separated by septa, the human duct has no such clear divisions, making functional analyses even more challenging. Over the past several years, we (Harris & Coleman, 1989; Pollard et al, 1991; Bischof et al, 2013; Browne et al, 2014, 2016a, 2016b, 2018, 2019; Leir et al, 2015), and others (Dube et al, 2007; Thimon et al, 2007; Cornwall, 2009; Belleannee et al, 2012a; Sullivan & Mieusset, 2016; Legare & Sullivan, 2019; Sullivan et al, 2019), have made a concerted effort to advance understanding of the human organ, to facilitate novel therapeutic approaches for male infertility and the development of targeted male contraceptives.

The human epididymis does not have an initial segment, rather the efferent ducts (EDs) provide the conduit from the testis to the head of the epididymis (caput) where the key functions of sperm maturation are thought to occur. Based on their gene expression profiles and other data, the corpus and cauda regions probably have a more important role in sperm storage and in ensuring the sterility of more proximal regions of the duct (Thimon et al, 2007; Belleannee et al, 2012b; Browne et al, 2018, 2019). Because of its dominant role in male fertility, we focused on the proximal part of the duct and generated a detailed single-cell atlas of the human caput epididymis, which is described here.

## Results

There is remarkable diversity in the structure of the epididymis from different donors as shown in Fig 1, making precise dissection of the caput tissue (in the absence of septa in humans) somewhat challenging. On the proximal side, our goal was to minimize the contribution of ED tissue and on the distal side to not include corpus tissue. It was not possible to take prospective tissue sections for histology from the same epididymis samples used to isolate single cells for single-cell RNA-sequencing (scRNA-seq) for reasons of speed and recovery of sufficient numbers of cells. Sections taken from EDs and proximal, mid, and distal caput tissue are shown in Fig S1A–D. However, having trained on more than 60 donor tissues (Leir et al, 2015; Browne et al, 2019), we were confident that we recovered primarily caput cells from the three donors used in the following scRNA-seq analysis. This was confirmed using our

Department of Genetics and Genome Sciences, Case Western Reserve University School of Medicine, and Case Comprehensive Cancer Center, Cleveland, OH, USA

Correspondence: ann.harris@case.edu

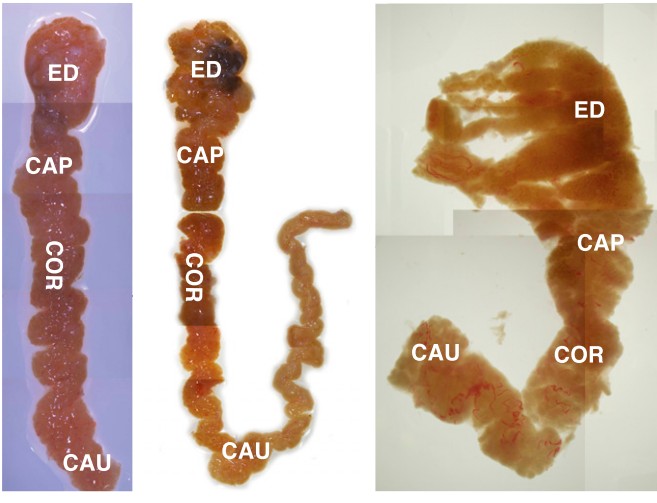

**Figure 1. Epididymis tissue from different donors shows substantial anatomical diversity, although single cells isolated from caput tissue from three donors show good correlation (Fig S1E).**
Images of dissected human epididymis tissue with efferent duct, caput (CAP), corpus (COR), and cauda (CAU) regions marked. Size bar = 1 cm.

previously published bulk RNA data from the caput, corpus, and cauda tissue (Browne et al, 2016b). We recovered 1,876, 1,309, and 2,114 cells from donors aged 31, 57, and 32 years, respectively, that passed quality control on the 10X Genomics Chromium System pipeline, providing scRNA-seq data on a pool of 5,299 cells. The cell–UMI matrix was normalized and clustered using Seurat version V3 (Satija et al, 2015). The clustering of cells from the three donors is shown in the Uniform Manifold Approximation and Projection (UMAP) dimension reduction plot (Becht et al, 2019) in Fig S1E, where eight distinct clusters are identified. Individual clusters contain cells from each donor (AXH009, AXH012, and AXH014), although as expected, the numerical contribution of cells varied by cluster (Table S1).

### Identification of major and minor cell populations in the human caput epididymis

Our current understanding of the cell types in the caput epididymis epithelium is largely based on careful characterization of the rodent tissue (reviewed in Breton et al [2016]) and some additional data from human tissues (reviewed in Sullivan et al [2019]). We expect to find principal cells as the most abundant cell type, with contributions from basal cells, clear cells, stromal cells, and other minor populations (reviewed in Sullivan et al [2019]). Each cell type has a set of identifying protein markers, which we used to guide our initial analysis. First, we examined the list of differentially expressed genes (DEGs) for each of the eight clusters identified in Seurat v3 (Satija et al, 2015) (Table S2) and shown by the UMAP plot in Fig 2A. A heat map illustrating the top 10 most DEGs in each of the 8 clusters shows the strong identity of each cell type (Fig 2B). The largest groups of cells in cluster 0 and cluster 1 have somewhat similar gene expression profiles based on their adjacent location on the UMAP plot and both clusters express some predicted markers of

principal cells. One classical marker of principal cells in the mammalian epididymis is the water channel aquaporin 9 (AQP9) (Da Silva et al, 2006; Hermo & Smith, 2011; Schimming et al, 2017), although our data suggest that its expression may be highest in a distinct population of principal cells (cluster 0) and much lower in cluster 1 cells (Fig S2A and B). Later, we present evidence that cluster 1 cells may encompass apical and narrow cells, rather than principal cells. The next most abundant cluster of cells (cluster 2) is primarily contributed by donors AXH012 and AXH014 and inspection of the DEG list suggests these may arise from ED cells contaminating the caput tissues. The diagnostic marker for the ED is villin (*VIL1*), which is apparently abundant in the ED epithelium but not seen in the caput epithelium in tissue sections (Legare & Sullivan, 2019; Sullivan et al, 2019). Of note, not all cells in this cluster express villin (Fig 2C), but the fact that they cluster suggest that the ED cell gene expression signature is substantially different from the other cells types in the caput tissue. Cluster 3 cells show markers of basal cells, which as their name implies are located as the base of the epididymis epithelium, although may exhibit processes into the lumen of the duct (Shum et al, 2008). The most differentially express gene in this cluster is keratin 5 (Fig 2C). Cluster 4 encompasses cells with high expression of mesenchymal markers and may include stromal cells, fibroblasts, and muscle cells. Cluster 5 cells are spermatozoa based upon high expression of sperm proteins such as cilia and flagella associated protein 43 (*CFAP43/WDR96*) and the FOXJ1 transcription factor, which is diagnostic of ciliated cells. This cluster is primarily contributed to by two of the tissue donors (AXH012 and AXH014). Cluster 6 cells are likely clear cells, which express the multiple subunits of the V–ATPase complex (Fig 2C), which is critical for the acidification of the epididymis lumen. A low pH in the lumen of the rodent epididymis is thought to be necessary for maintaining sperm quiescence (Breton et al, 1996; Breton & Brown, 2013). Cluster 7 cells are clearly immune cells with multiple HLA peptides differentially expressed, suggestive of B-cells and the Fc Fragment of IgE Receptor Ig (*FCER1G*) gene, which encodes the Fc ε receptor that is strongly associated with monocytes and macrophages. It is possible that this cluster also includes the halo cells described in other species (Serre & Robaire, 1999).

### Regional and cell-specific distribution of function in the major caput epithelial cell populations

Principal cells are thought to be most abundant cells throughout the length of the epididymis and are predicted to have some regional specialization of function between the caput, corpus, and cauda regions. We observed earlier that the major cell type in cultured epithelial cells derived from these regions had some characteristics of principal cells, but caput-derived cells had a substantially different morphology from the corpus and cauda cells (Leir et al, 2015). Hence, it is of interest here to observe a clear functional specialization of the predicted principal cells within the caput (clusters 0 and 1). The most differentially over-expressed genes in cluster 0 include those encoding multiple antimicrobial peptides, including β defensins (DEFB118 [Fig S2C and D], 119 [Fig 2C], 121 and 123, all of which map to the 20q11 defensin gene cluster) and DEFB128 (at 20p13), and sperm-associated antigen 11A and 11B (SPAG11A/B [Fig 3A–F]). The B form of SPAG11 contains a C-terminal

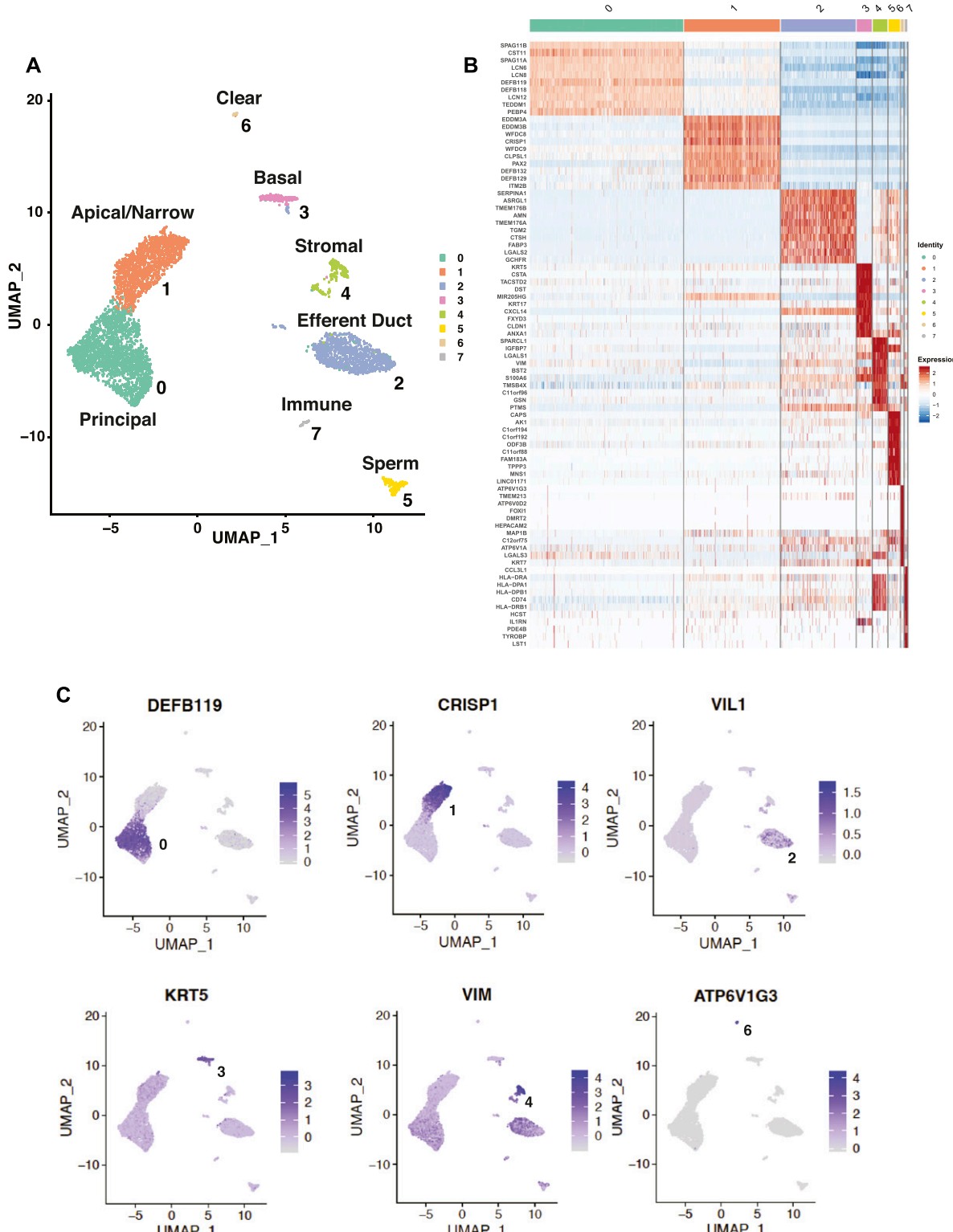

**Figure 2. Eight distinct cell types are identified in the proximal epididymis.**
**(A)** UMAP dimension reduction plot of scRNA seq data from donors AXH009, 12, and 14 combined and clustered according to gene expression profiles. Based on differentially expressed genes (DEGs) in each cluster (Table S2), the 8 clusters identified are 0, principal cells; 1, apical and narrow cells; 2, efferent duct cells; 3, basal cells; 4 stromal/muscle cells; 5, spermatozoa; 6, clear cells; and 7, immune cells. **(B)** Heat map showing top 10 most DEGs in each cluster ($Log_2$). Each row represents a gene and each column a cell in each cluster (identity) with the color intensity corresponding to expression levels. **(C)** UMAP dimension reduction plots to show expression of key marker genes in each cluster, with DEGs colored according to the $Log_2$ scale shown on the right of each panel. Cluster 0 *DEFB119*; Cluster 1 *CRISP1*; Cluster 2 *VIL1*; Cluster 3 *KRT5*, Cluster 4 *VIM*; and Cluster 6 *ATP6V1G3*.

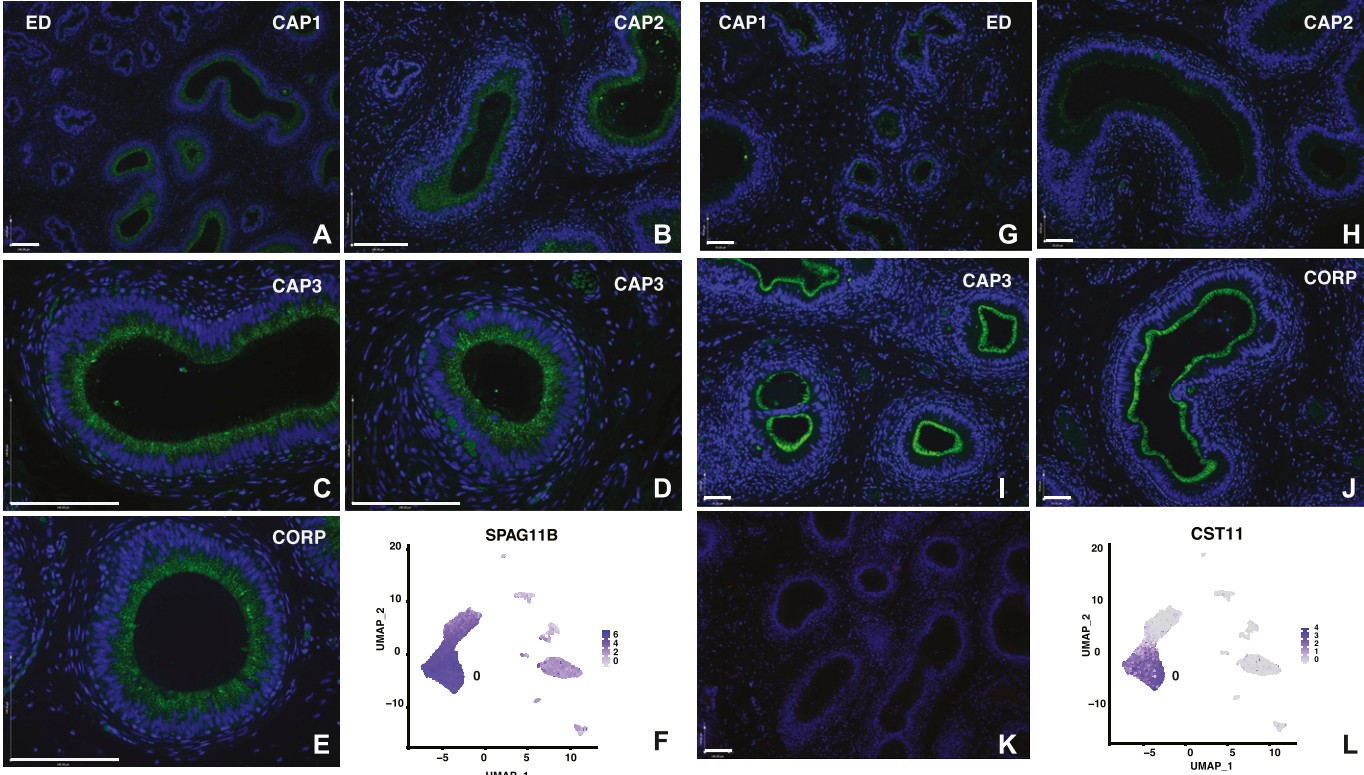

**Figure 3.   Cell-specific expression of key genes in the caput epididymis cell clusters: *SPAG11B* and *CST11* in principal cells.**
(i) Expression of SPAG11B **(A, B, C, D, E)** by immunofluorescence and scRNA-seq UMAP plot **(F)**. SPAG11B is evident in cluster 0, 1, and 2 group cells and is a strong differentially expressed gene in principal cells (0). Although the protein is at very low levels in the efferent ducts (A), principal cells in sequential caput regions (B, C, D) and the corpus (E) show similar abundance. Size bar = 140 μm. (ii) Expression of CST11 **(G, H, I, J)** by immunofluorescence and scRNA-seq UMAP plot **(L)**. CST11 is evident primarily in cluster 0 principal cells but is regionally distributed along the genital duct with no staining in efferent ducts, donor dependent staining that is high or low in proximal caput, consistently low in mid-caput and high in distal caput and corpus. **(K)** No first antibody control. (G, H, I, J), size bar = 70 μm, K = 140 μm. Abbreviations: Efferent ducts, ED; Sequential caput regions, CAP1-3; Corpus, CORP.

β-defensin domain which is lacking from the A form (Ribeiro et al, 2015). Also abundant in cluster 0 cells are transcripts for cystatin 11 (*CST11*) a type 2 cysteine protease inhibitor (Hamil et al, 2002) (Fig 3G–L). CSTs have multiple roles in the mammalian reproductive tract including as antimicrobial agents. Adhesion G protein-coupled receptor G2 (*ADGR2* also known GPR64*) and the serine protease inhibitor serpin family F member 2 (*SERPINF2*), which are both androgen-responsive genes in human epididymis epithelial cells (Yang et al, 2018), are also significantly over-expressed in cluster 0 cells. Also of note in this cluster are several lipocalins (*LCN6, 8,* and *12*), LCN8 is reported to be an epididymis specific lipocalin (Suzuki et al, 2007).

In contrast, cluster 1 cells show differential expression of a different group of well characterized caput epididymis markers including epididymal protein 3A and 3B encoded by (*EDDM3A/3B*) and cysteine-rich secretory protein 1 (*CRISP1*) (Figs 2C and 4A–F and K). CRISP1 is known to be secreted by the epididymis epithelium into to the lumen where it binds to the sperm head and has a role in sperm–egg fusion (reviewed in Evans [2002]). Also abundant in cluster 1 cells are whey acidic protein four-disulfide core domain protein 8 and 9 (*WFDC8/9*) (WFDC8 is shown in Fig 4G, H, and K, with its regional distribution shown in Fig S3A–D). WFDC8/9 are epididymis protease inhibitors (Rajesh et al, 2011). Transcripts from the Paired Box gene 2 transcription factor gene (*PAX2*), which is known to direct a transcriptional network in

urogenital cells including the epididymis cells (Browne et al, 2018, 2019) is also abundant in cluster 1. Similarly over-expressed in cluster 1 cells are a group of β defensins. However, these are encoded by different genes than those seen in cluster 0 cells and include *DEFB132*, *DEFB129*, and *DEFB127*, which map to the defensin gene cluster on chromosome 20p13, and DEFB110, DEFB134, and DEFB131, which are encoded by defensin gene clusters on three different chromosomes (6, 8, and 4, respectively).

We subclustered groups 0 and 1 alone using DIM = 10 and DIM = 30, in an attempt to reveal the differences in their identity, but this was not highly informative: a small cluster of cells, arising primarily from one tissue donor, are likely adipocytes based upon differential expression of adipogenesis regulatory factor (*ADIRF*) (data not shown). In an effort to determine whether the cluster 0 and cluster 1 cells represented principal cells with significantly different functions based upon their regional localization, or whether they identified different cell types in the caput epithelium, we used immunofluorescence to examine the expression pattern of proteins encoded by cluster-specific DEGs. Sequential panels in Fig 3A–E and G–J show localization of SPAG11B and CST11 (cluster 0 markers), respectively, from the EDs through the proximal (CAP1), mid (CAP2), and distal (CAP3) caput and Fig 4A–D illustrates CRISP1 (cluster 1 marker) protein in parallel sections.

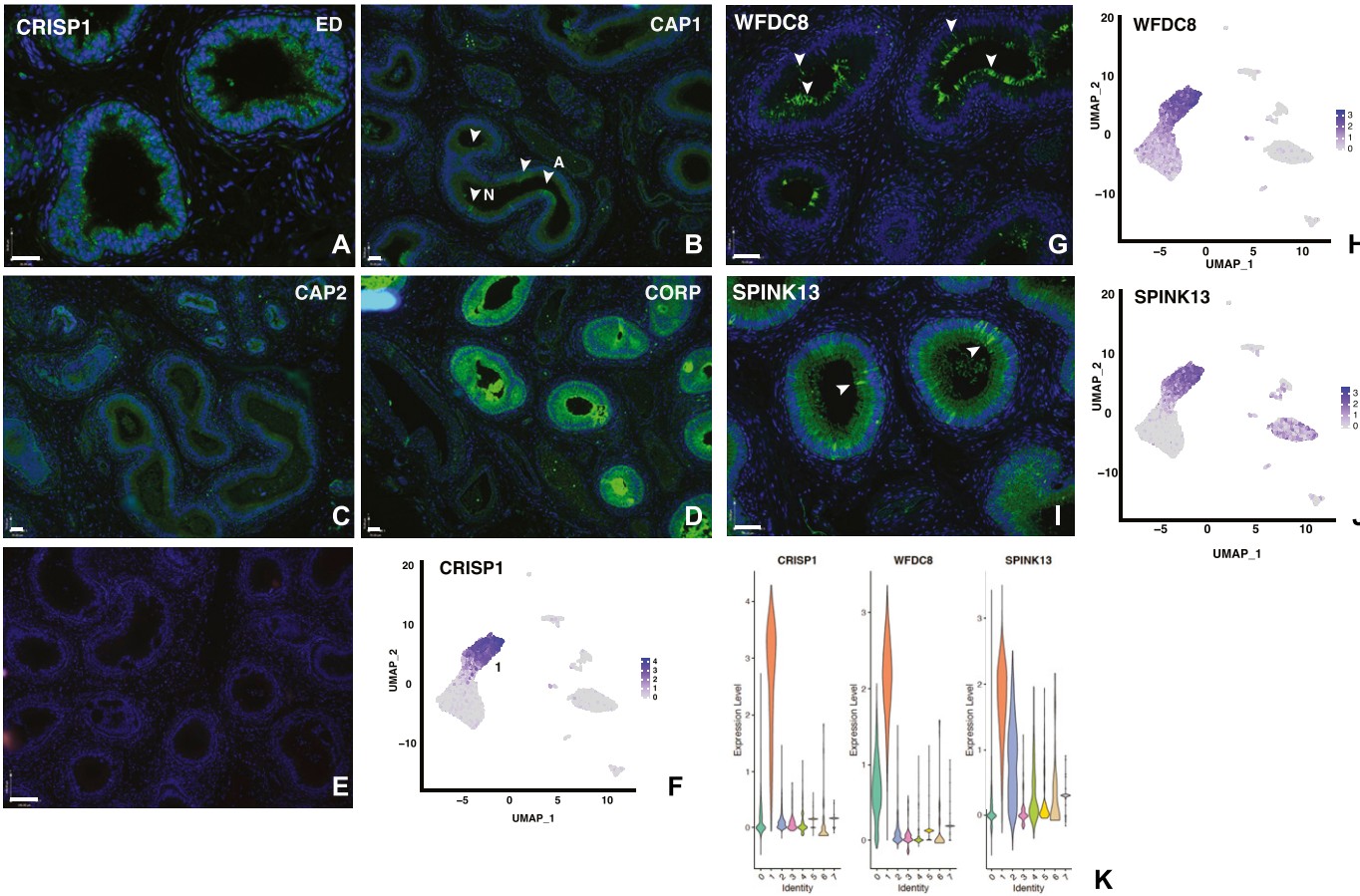

**Figure 4. Cell-specific and regional expression of key genes in the caput epididymis cell clusters: *CRISP1*, *WFDC8*, and *SPINK13* in apical and narrow cells.**
(i) Expression of CRISP1 by immunofluorescence **(A, B, C, D)**, no first antibody control **(E)** and scRNA-seq UMAP plot **(F)**. CRISP1 is evident primarily in cluster 1, apical, and narrow cells, but is regionally distributed along the genital duct with notable staining in the efferent ducts and proximal caput, low in mid-caput and very high in the corpus. Arrows in (B) highlight patches of staining in apical (A) cells at the epithelial surface and rare narrow (N) cells. Size bar = 36 $\mu$m (A) 70 $\mu$m (B, C, D), =140 $\mu$m (E). (ii) **(G, H)** Expression of WFDC8 (G) by immunofluorescence with corresponding UMAP plot showing high expression in cluster 1 (H). **(I, J)** Expression SPINK13 (I) by immunofluorescence with corresponding UMAP plot, showing high expression in cluster 1 (J). Size bar = 70 $\mu$m all panels. **(K)** Violin plots showing cell-specific expression cluster 1 markers, *CRISP1*, *WFDC8*, and *SPINK13*.

SPAG11B is not expressed in the ED epithelium (Fig 3A) but is seen in most cells throughout the caput (Fig 3B–D) through the corpus (Fig 3E) at approximately the same abundance. Of note, although *SPAG11B* is the top DEG in cluster 0 cells (Table S2), it is also expressed at a lower level in cluster 1 cells (Fig 3F), consistent with SPAG11B protein detection in nearly all surface cells in the caput epithelium. In contrast, CST11 protein, although also absent from the ED (Fig 3G), shows a gradient of expression along the caput. It is seen at low levels in the proximal caput close to the ED region (Fig 3G), although it is almost undetectable in the adjacent caput region (Fig 3H). CST11 is much more abundant in the middle of the caput (Fig 3I) and through the corpus (Fig 3J). However, some variation was noted between different tissue donors. CRISP1 shows a markedly different pattern of expression (Fig 4) where moderate expression levels are seen in the ED (Fig 4A) and caput epithelium through all regions (Fig 4B and C), although the protein is much more abundant in the corpus (Fig 4D). Careful inspection of the CRISP1 localization in the proximal and mid-caput shows much higher expression in what appear to be patches of surface/apical epithelial cells and in rarer narrow cells (marked by arrowheads [A and N, respectively] in Fig 4B). Of

note, the distribution of WFDC8 (cluster 1 marker) in Fig 4G coincides with the patches of the surface (apical) epithelial cells, whereas immunofluorescence detection of serine peptidase inhibitor Kazal type 13 (SPINK13), another DEG in cluster 1 cells, highlights narrow cells (Fig 4I, white arrowheads, Fig 4J). Higher resolution images of these cell types are shown in Fig S3E and F. These data suggest that cluster 0 may primarily encompass principal cells, whereas cluster 1 includes other cell types such as apical and narrow cells. Moreover, the results support a model whereby both the regional expression of markers along the caput within a single cell type/cluster and cell cluster–specific markers combine to integrate the regional functions of the epididymis.

## The unique identity of efferent duct epithelial cells

Since there are no septa dividing the different functional regions of the human epididymis and our goal was to capture all cell types in the caput region, we were not surprised to detect some ED cells in our single cell preparations. These cells (cluster 2) were mainly

contributed by two donor samples. We identified the ED cells primarily by the expression of villin (VIL1) (Fig 5A and E), which is expressed in the apical membranes of ED but not caput epididymis epithelium (Legare & Sullivan, 2019; Sullivan et al, 2019). Not all cells in cluster 2 express high levels of villin (Fig 5E), suggesting that only a subgroup of surface epithelial cell express the marker. However, the clustering suggests that other ED cell types are more similar to each other than to the caput-derived cells in other clusters (e.g., 0 and 1). Other abundant transcripts in ED cells include asparaginase and isoaspartyl peptidase 1 (ASRGL1) a gene associated with prostate cancer (Pudova et al, 2019), the estrogen receptor (ESR1, Fig 5C) and FXYD domain containing ion transport regulator 2 (FXYD2) that encodes the sodium/potassium-transporting ATPase subunit γ, which is involved in renal transport (Sha et al, 2008) (Fig 5G). Using antibodies specific to the estrogen receptor (ESR1) and the androgen receptor (AR) in immunofluorescence we confirmed that ESR1 levels were high in the ED epithelium (both nuclear and cytoplasmic) (Fig 5C) but absent from the caput epithelium, where AR was clearly localized to the nuclei (Fig S3G–I). By re-clustering cells in cluster 2 with a default DIM of 10, we defined three main cell types in the ED cell population (Fig S4A). The cell types within the human EDs are not well documented (reviewed in Hess [2002]). Based upon DEG lists (Fig S4B) and comparison with the transcriptional signatures of different cell types in the epididymis these cells are predicted to be principal cell-like (Group 0), basal cell-like among others (Group 1), and apical and narrow cell-like (Group 2). Reclustering with a DIM value of 24 identified 4 cell clusters, providing a separate identity to a small subset of basal cell (arrowed in Fig S4A), although their transcriptional signature alone did not enable definitive assignment of a differentiated cell type.

## Basal cells may include the stem cells of the epididymis epithelium

A relatively small number of predicted basal cells were recovered in the scRNA-seq (cluster 3), which is perhaps not unexpected as they form a discrete single layer of cells at the base of the epithelium (Fig 6). In earlier work on the rat epididymis, basal cells accounted for fewer than 10% of cells (Trasler et al, 1988). Despite this low number of cells, their identity is quite unique. In addition to the classical marker of basal cells in many epithelia, keratin 5 (KRT5) (Leir et al, 2020) (Fig 6A and C), which is the most DEG, the list of the top 25 DEGs show many that encode proteins involved in the cytoskeleton and extracellular matrix (Table S2). Among these are claudin 1 (CLD1) (Gregory et al, 2001), fibronectin (FN1), integrin alpha 2 (ITGA2), LIM domain and actin-binding protein 1 (LIMA1), dystonin (DST), and keratin 17 (KRT17) (Fig 6F). Also on the list are other proteins that may reflect functions of the basal cells that protrude from the base of the epithelium through to the lumen, as observed in other species (Shum et al, 2008) and illustrated in Fig 6A, yellow arrowhead. These may account for the high expression of FXYD3, which is thought to regulate ion pumps and channels (Crambert et al, 2005), and tumor-associated calcium signaling transducer 2 (TACSTD2), a cell surface receptor transducing calcium signals (Nakatsukasa et al, 2010). Also of note in cluster 3 are abundant transcripts from the TP63 gene (Fig 6A, D, and E), which encodes the p63 transcription factor, known as a marker of stem/progenitor

cells in other epithelia such as the airway (Zuo et al, 2015). Immunolocalization of p63 in the caput epididymis clearly showed high abundance in only a subset of KRT5 positive basal cells (Fig 6A, purple arrow highlighting purple cells). This predicts that, as has been suggested elsewhere (Pinel et al, 2019), basal cells may be the source of stem cells for regeneration of the epididymis epithelium.

## Interstitial cells of the caput epididymis

Careful dissection of the epididymis before single-cell sequencing requires the removal of substantial amounts of connective tissue and often adipose tissue deposits to reveal the tubular structure. Hence, it is not surprising that we recover the stromal cell types identified in cluster 4, although some of these may also be an integral part of the epididymis duct structure. Notable among DEGs in this group (Table S2) are vimentin (VIM) (Fig 2C), encoding the cytoskeletal intermediate filament protein found in non-epithelial cells such as mesenchymal cells, bone marrow stromal cell antigen 2 (BST2) and myosin light chain 9 (MYL9), a myosin regulatory subunit with a role in both smooth muscle and non-muscle cells. Other DEGs in this cluster include a number that encode proteins involved in actin filament polymerization and smooth muscle biology, for example, thymosin beta 4 X-linked (TMSB4X), gelsolin (GSN), and transgelin (TAGLN). Differential expression of matrix gla protein (MGP) and SPARC-like protein 1 (SPARCL1) also indicates that this cluster of cells may include blood vessel derivatives.

## Spermatozoa

In our previous analysis of gene expression in the epididymis, we detected signatures that appeared to be from sperm (Browne et al, 2016b); hence, the identification of cluster 5 as sperm, based upon on their DEGs (Table S2), was expected. At the top of the DEG list is calcyphosine (CAPS), which encodes a calcium-binding protein with a predicted role in regulating ion transport. The gene was also named epididymis secretory sperm binding protein implicating a source exogenous to sperm. Also among the DEGs is adenylate kinase (AK1), which has a key role in energy metabolism, consistent with the motility requirements of sperm, and cation channel sperm associated auxiliary subunit delta (CATSPERD) a component of the CatSper complex involved in hyperactivation of sperm, which is required for sperm mobility. Another DEG with a predicted role in sperm motility role is tubulin polymerization promoting protein family member 3 (TPPP3), the regulator of microtubule dynamics and bundling. Other DEGs with a role in the structure of motile cilia include radial spoke head component 1 (RSPH1), the primary cilia formation (PIFO) gene, which has a role in cilia disassembly, multiple coiled-coil domain-containing protein genes such as cilia and flagella-associated protein 53 (CFAP53/formerly CCDC11) and 43 (CFAP43, also known as WDR96). Consistent with the importance of cilia in these cells is the identification of the forkhead box J1 (FOXJ1) transcription factor as a DEG. Also relevant to sperm function is dynein light chain roadblock-type 2 (DYNLRB2), which encodes an accessory component of the cellular motor dynein that facilitates movement of cargo along intracellular microtubules. In addition to novel sperm-associated genes in cluster 5 cells, known markers of spermatogenesis were also identified as

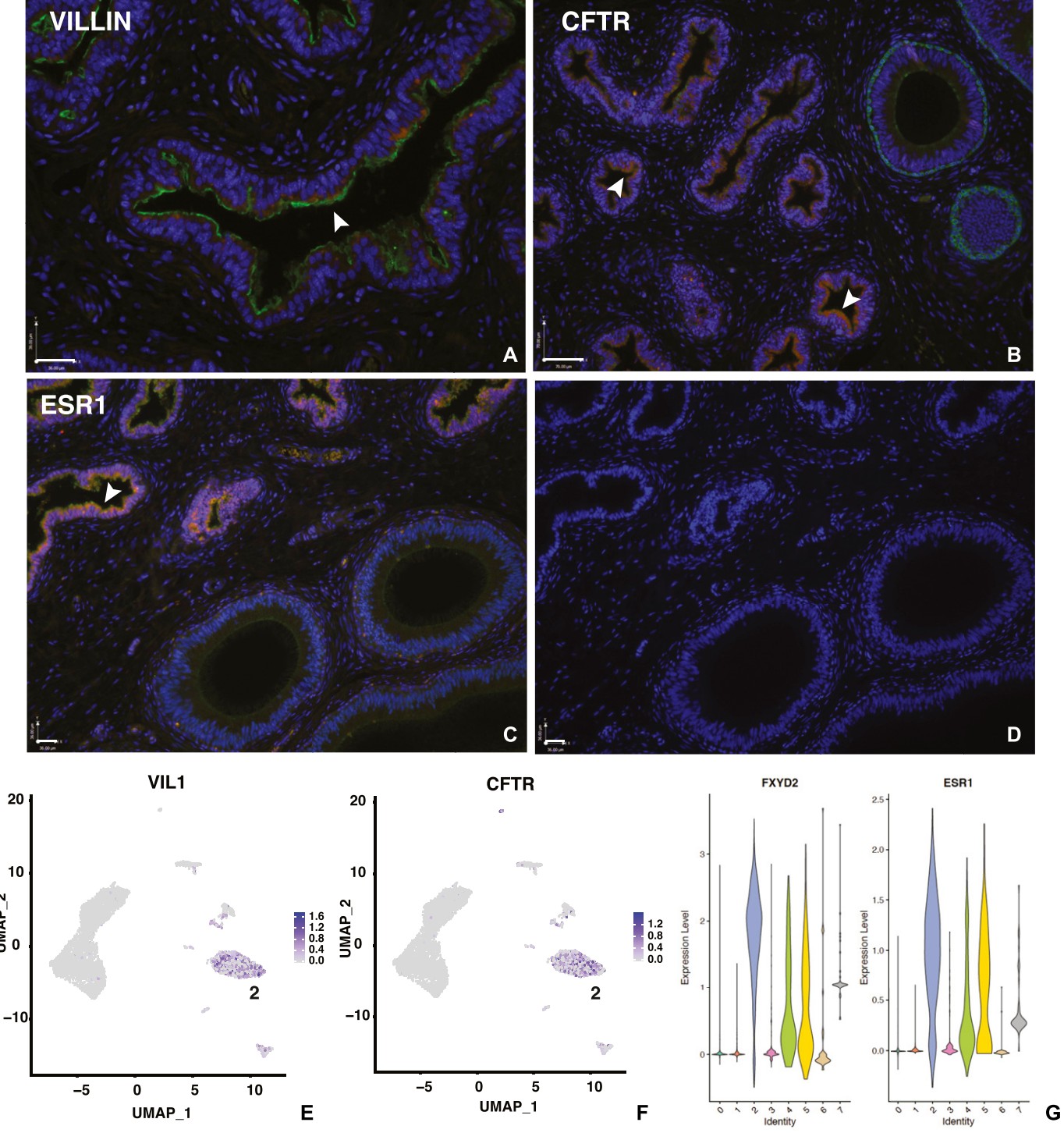

**Figure 5. Cell-specific expression of key genes in the efferent duct: *VIL1*, *CFTR*, and *ESR1*.**
**(A, B, C)** Expression of VIL1 at epithelial surface (A, green), CFTR, both cytoplasmic and surface staining (B, red; basal cells are marked by KRT5, green in adjacent caput), and ESR1, both nuclear and cytoplasmic (C, red). **(D)** No first antibody control. Size bar = 36 μm panels (A, C, D), 70 μm in (B). **(E, F)** UMAP plots showing *VIL1* and *CFTR* expression each in a subset of cluster 2 cells, some overlapping. **(G)** Violin plots showing high expression of *FXYD2* and *ESR1* in cluster 2 cells.

DEGs, such as meiosis-specific nuclear structural 1 (*MNS1*) and rhophilin-associated tail protein 1 like (*ROPN1L*). A violin plot of gene expression by cluster for several key DEGs in cluster 5 is shown in Fig S5A.

**Clear cells: the epididymis ionocyte**

Clear cells in the epididymis are known to express high levels of the vacuolar ATPase (V-ATPase), which pumps hydrogen ions into the

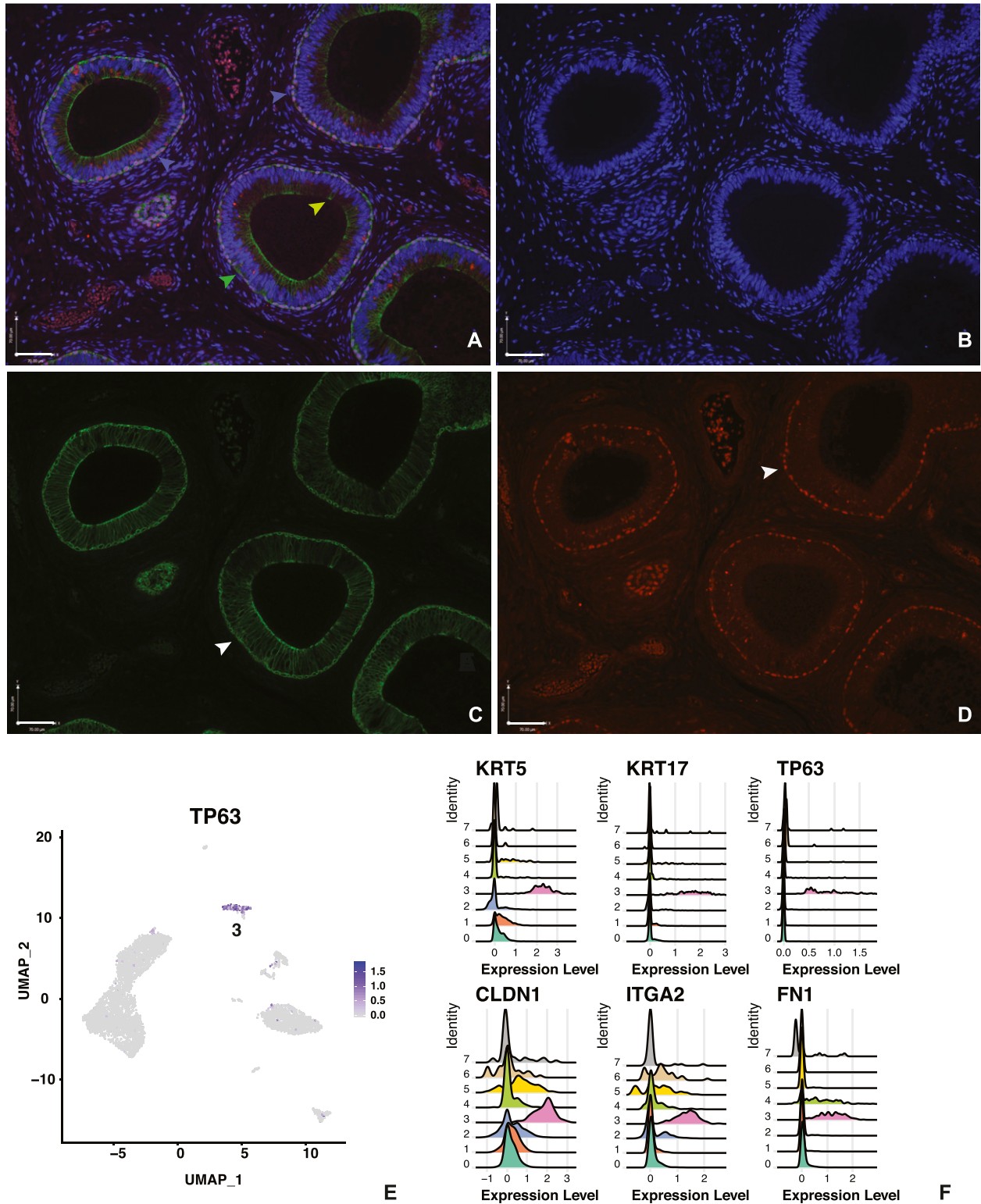

**Figure 6. Cell-specific expression of key genes in the caput epididymis cell clusters: *KRT5* and *TP53* in basal cells.**
**(A, C)** Expression of KRT5 in cluster 3, basal cells (A, C, green) both surrounding the caput duct (green arrowhead) and in rarer cells protruding to the epithelial surface (yellow arrowhead). **(B)** DAPI staining of nuclei alone. **(A, D)** Expression of TP63 (A, D, Red) in a subpopulation of basal cells. Purple cells in (A) marked by purple arrowheads show cells staining for both KRT5 and TP63. Size bar = 70 μm all panels. **(E)** UMAP plot showing TP63 expression in cluster 3 cells. **(F)** Ridge plots showing expression of *KRT5*, *KRT17*, *TP63*, *CLDN1*, *ITGA2*, and *FN1* predominantly in cluster 3 cells.

lumen of the duct, and thus have a pivotal role in maintenance of luminal pH in the mouse (Breton et al, 1996; Shum et al, 2009; Park et al, 2017). Cross talk between clear cells and principal cells, which express the sodium/hydrogen exchanger (NHE3) and the CFTR (Leir et al, 2015; Park et al, 2017) is thought to coordinate the luminal pH that is required for sperm quiescence. However, recent scRNA-seq of the lung epithelium (Montoro et al, 2018; Plasschaert et al, 2018) defined the rare high *CFTR*-expressing cells that were observed earlier in several epithelia (Trezise et al, 1993; Engelhardt et al, 1994; Ameen et al, 1995) as pulmonary ionocytes and showed them to contain multiple ion channels and high levels of the transcription factor FOXI1. With these observations in mind, we examined the DEGs in cluster 6 cells (Table S2) and found them to have characteristics of both clear cells and ionocytes. For example, at the top of the DEG list are several subunits of the vacuolar ATPase, including cytosolic ATPase H+ transporting V1 subunits G3, A and B (*ATP6V1G3* [Fig 7A and C], *ATP6V1A*, and *ATP6V1B1*) and transmembrane ATPase H+ transporting V0 subunit D2 (*ATP6V0D2*) and A4 (*ATP6V0A4*). This suggests cluster 6 includes clear cells. However, also among the most significant DEGs is *FOXI1* (Fig 7G) and though of lower significance, but still among DEGs, are both *CFTR* (Fig 7F) and the gene encoding the α subunit of the epithelial sodium channel (*SCNN1A*). These data are consistent with cluster 6 cells being equivalent to the "ionocytes" of the epididymis. Hence, clear cells and ionocytes are likely the same cell type in the male genital duct. Of note, there are many other DEGs in cluster 6, some with known functions including several transcription factors, and many with functions previously not associated with differentiated clear cells. Both the scRNA-seq data (Fig 7E) and further immunofluorescence imaging (data not shown) demonstrate that only a subset of V-ATPase expressing cells also show CFTR protein. This was confirmed at the RNA level in the accompanying feature scatter plot (Fig S5B and C). It is possible that the histological definition of a clear cell encompasses more than one cell type, which could account for the diversity in this cluster. An extensive functional analysis is warranted to resolve the precise function of these cells in epididymis epithelial biology.

### Immune cells in the epididymis

We showed data above that demonstrated a key contribution of genes and pathways of innate immunity to principal cells in groups 0 and 1 in particular. Now considering adaptive immunity, we identify most immune cells in cluster 7. At the top of the DEG list (Table S2) are several HLA molecules including Class II MHC molecules *HLA-DRA.1*, *DPA1.1*, *DPB1.1*, *DRB1.1*, and *DQB1.1* indicative of antigen-presenting cells. Based on additional DEGs, including CD22, which mediates B-cell interactions, together with the absence of T-lymphocyte–specific DEGs suggests the cluster contains a majority of B cells. Also present are macrophages/monocytes as evidenced by the DEG Fc fragment of IgE receptor Ig (*FCER1G*), which is a macrophage/monocyte marker not expressed on B cells. Many other immune-related genes are differentially expressed in cluster 7 cells. Of note, the macrophage-like cells described previously in the epididymis of rodents are Halo cells (Serre & Robaire, 1999), which are a low abundance cell type that likely contribute to the

pool in cluster 7. A feature dot plot showing key marker gene expression changes across clusters 0–7 is shown in Fig 8.

### CFTR and its role in the epididymis

CFTR, a small conductance anion channel, has a pivotal role in normal epididymal fluid transport. Loss-of-function mutations in *CFTR* are associated with absence of the vas deferens and epididymis abnormalities in cystic fibrosis (CF) (Landing et al, 1969; Holsclaw et al, 1971) and abundant CFTR expression is seen in the genital duct epithelium of humans and many other species (Harris & Coleman, 1989; Harris et al, 1991; Pollard et al, 1991; Bertog et al, 2000; Leung et al, 2001). Very high levels of *CFTR* expression are seen only in a subset of cells in cluster 6, clear cells/epididymis ionocytes (Fig 7B and F), while *CFTR* is also a DEG in some cells within the ED cluster 2 (Fig 5B and F). However, of note, the cluster 2 cells likely include more than one cell type, since re-clustering based on expression of villin and CFTR identified 222 cells that expressed only VIL1, 150 only CFTR, and 150 cells both VIL1 and CFTR (Fig S4C). Few cells in the principal (cluster 1), apical/narrow (cluster 1), basal (cluster 3), stromal (cluster 4) cell clusters, and sperm (cluster 5) show high expression of CFTR, although most cells in these clusters express low levels of the gene. This observation is in contrast to data from cultured HEE caput cells, which were thought to be most similar to principal cells, but where CFTR is more abundant than in tissue-resident principal cells (Leir et al, 2015; Browne et al, 2016b). These differences may result from the cultures conditions, including lack of relevant cell:cell cross-talk, altered substrate and culture media causing a loss or change of cell identity.

## Discussion

The generation of single cell sequencing data from human tissues is transforming our understanding of biological mechanisms. This is particularly true for poorly studied organs and tissues that are difficult to obtain. One such tissue is the human male genital duct, the epididymis and vas deferens, which have a pivotal role in sperm maturation and hence maintenance of the species. Most models of epididymis function are based upon other mammals, which show substantial anatomic and functional diversity. Here, we construct a single cell atlas of the human proximal epididymis, which reveals detailed molecular characterization of both common and rare cell types, and hence may advance our understanding of mechanisms of male fertility.

The key functions of the epididymis are performed by the cells lining the lumen of the duct, which maintain a low pH environment necessary for sperm quiescence. These cells also secrete a wide spectrum of proteins, peptides, and RNAs that provide the necessary cues for normal sperm maturation and prevent damage by external stimuli such as infections. However, the precise identity of the cellular origin of many of these key components is not clear. Our data show a marked regional gradient of function along the proximal epididymis, which is supported by specific groups of individual epithelial cell types. Principal cells are known to be the majority cell type in the epididymis epithelium and predominate in most epididymis cell culture models. It has been suggested that

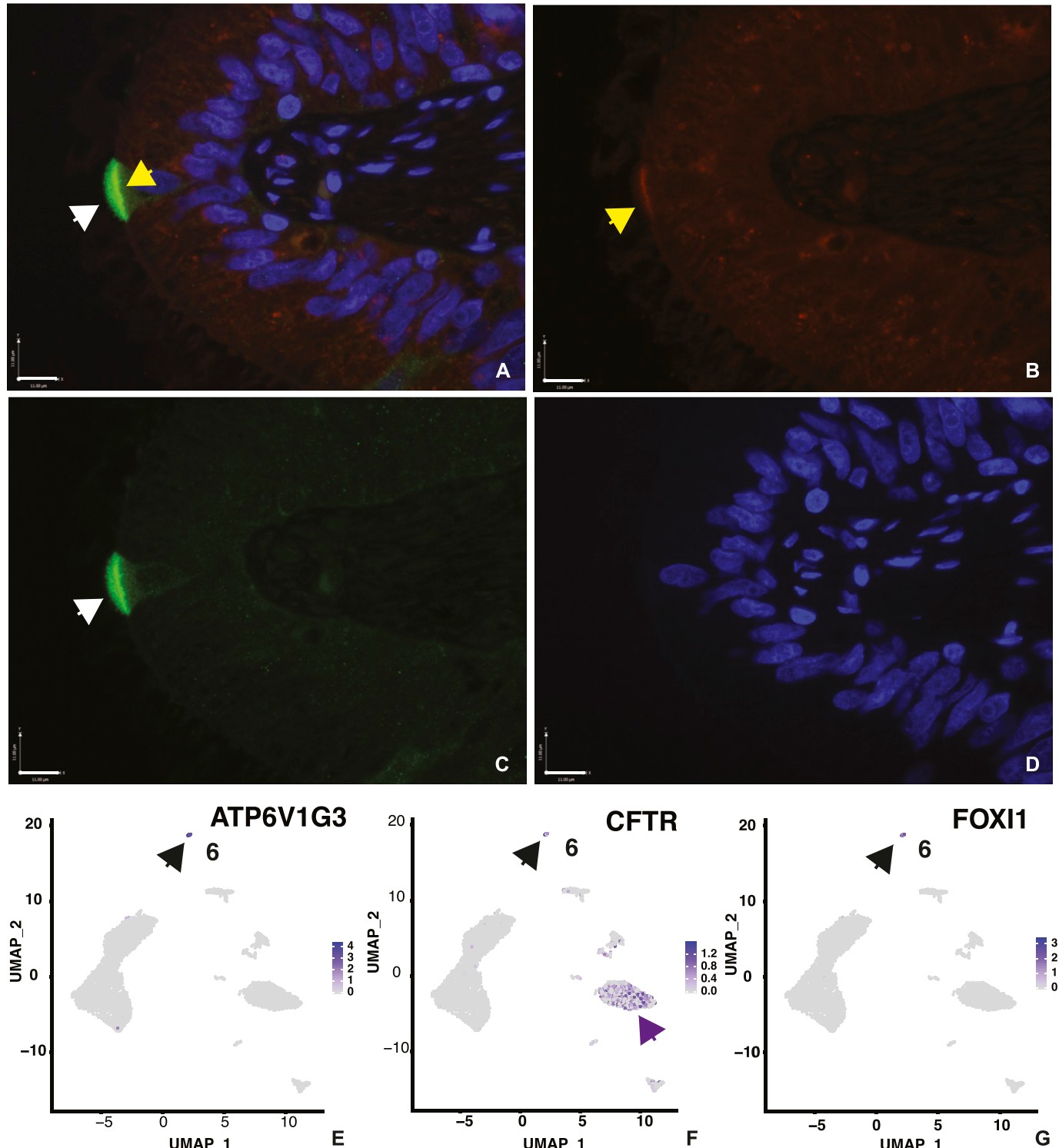

**Figure 7. Cell-specific expression of key genes in the caput epididymis cell clusters: *ATP6V1G3* and *CFTR* in clear cells.**
**(A, B, C)** Expression of the V-ATPase subunit ATP6V1G3 (A and C green, marked by white arrows) and CFTR (A and B, red, marked by yellow arrows) in cluster 6 cells. Both proteins together appear as faint yellow in panel (A). **(D)** DAPI stain show nuclei. Size bar = 11 μm all panels. **(E, F, G)** are UMAP plots showing *ATP6V1G3*, *CFTR*, and *FOXI1* gene expression, respectively, in cluster 6 cells.

principal cells in the caput, corpus, and cauda epididymis and the vas deferens have different functions (Jervis & Robaire, 2001; Cornwall, 2009; Domeniconi et al, 2016), although supporting evidence is derived largely from bulk microarray or RNA-seq studies and studies of individual cellular processes. Here, we show by scRNA-seq and accompanying immunocytochemistry of tissue sections, that although

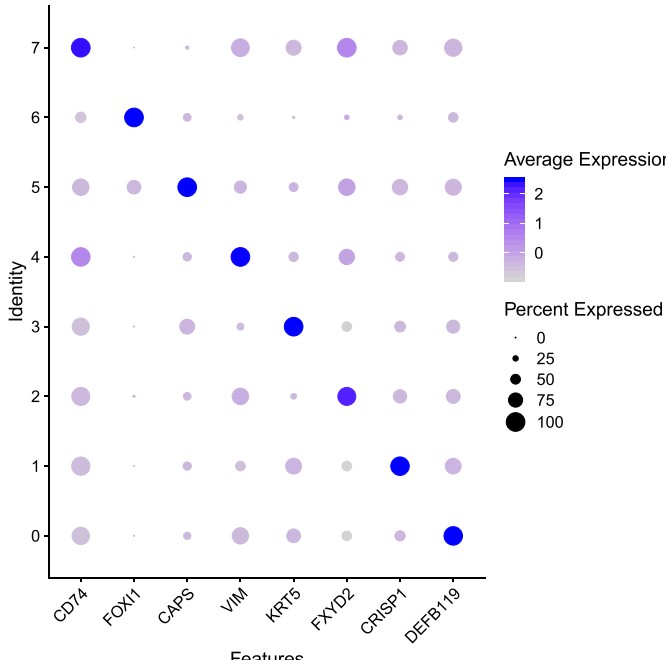

**Figure 8. Feature dot plot highlighting the expression of gene markers for each cell cluster.**
The size of the dot corresponds to the percentage of cells expressing the feature in each cluster, and its color intensity denotes the average expression level.

there is a profound regional distribution of expression of specific genes along the length of the proximal epididymis epithelium, this may not correlate with the localization of unique cell populations. Recent scRNA-seq data from the mouse genital duct deposited on BioRxiv (Shi et al, 2020 *Preprint*; Rinaldi et al, 2020 *Preprint*) are not directly comparable with our analysis, as they include all regions of the duct from the initial segment through the cauda epididymis or vas deferens, respectively, and hence identify many more cell populations.

First of note are the two most abundant groups of cells (clusters 0 and 1), which we initially identified as putative principal cells in the caput epithelium. Cluster 0 cells have a major function in producing antimicrobial peptides, including β defensins expressed from diverse gene clusters, and other components of the innate immune system. The β defensin family members *DEFB118* and *DEFB119* are among the most DEGs in group 0 cells, as are *SPAG11A* and *SPAG11B*, which have structural similarities to the β defensins. The expression of SPAG11B is seen in principal cells along the whole length of the caput by immunofluorescence. In addition to the antimicrobial activity of the encoded proteins/peptides, they are thought to have other functions and may be directly involved in sperm maturation (Pujianto et al, 2013). SPAG11B is widely distributed in the epididymis fluid in rodents where it also coats sperm; hence, it is of particular interest to identify its source in principal cells throughout the human caput epididymis epithelium. Other antimicrobial DEGs in cluster 0 cells are the cystatin cysteine protease inhibitors *CST11* and *CST3*. However, unlike SPAG11B, CST11 protein shows a marked regional distribution by immunofluorescence, with highest levels in the mid-to-distal caput epithelium and lower levels in the proximal portion. CST11 is the predominant

cystatin in the male reproductive tract of *Macaca mulatta* monkeys (Hamil et al, 2002), and like some other *SPAG* genes (Ribeiro et al, 2017), it is known to be regulated by androgens. Although the AR is not a DEG in cluster 0 cells, we show by immunofluorescence that AR is an abundant protein in the nuclei of epithelial cells lining the caput. The lipocalins 6, 8, and 12 are also differentially expressed in cluster 0 cells, and these proteins have a known role in transporting small hydrophobic ligands in their cup-shaped binding pocket (calyx) and transporting these to target cells. LCN8 is one of several epididymis-restricted lipocalins (Suzuki et al, 2007), suggesting that these may have a key role in sperm maturation. Cluster 1 cells have a less distinctive antimicrobial/defense gene expression signature than those in cluster 0: although other members of the β defensin gene family are among DEGs (*DEFB129*, *DEFB13*, and *DEF112*). Predominant transcripts in these cells were initially thought to reflect the secreted proteome of principal cells in the epididymis. CRISP1 is an androgen-responsive, abundantly secreted protein in the epididymis fluid, where it coats spermatozoa and is involved in both capacitation and fertilization (Ernesto et al, 2015). Hence, it was of interest to identify CRISP1 as a prominent DEG in cluster 1 cells, suggesting these may be the main source of this protein in the epididymis. However, our immunofluorescence data suggest that CRISP1 is localized in surface patches resembling apical cells and in a few narrow cells in the caput epithelium, unlike the distribution of principal cells. This is in marked contrast to its very abundant expression of CRISP1 in most cells in the corpus epithelium and in luminal secretions. To further investigate the possibility that cluster 1 cells encompassed a population of apical and narrow cells, we looked for markers of these two cell types in the DEG list. Apical and narrow cells were carefully examined in earlier work in the rat epididymis (Adamali & Hermo, 1996). Cathepsin D (CTSD) was highly expressed in both cell types, whereas a subunit of glutathione S-transferase (Y *f*, GST-P) was abundant only in apical cells. In contrast, β-hexosaminidase A was expressed at high levels in narrow cells, which were also the site of carbonic anhydrase II expression (CA2), suggesting a role in modifying the luminal pH. Among DEGs in cluster 1, we identified the genes encoding the beta subunit of hexosaminidase (*HEXB*) and carbonic anhydrase 8 (*CA8*), indicative of narrow cells and also gluthatione S-transferase Mu3 (*GSTM3*), suggesting the presence of apical cells. However, the presence of these markers in cluster 1 cells does not necessarily imply these cells have identical functions to the apical and narrow cells defined in the rodent initial segment and intermediate zone. The abundance of these putative apical and narrow cell types in the scRNA-seq data is somewhat surprising, given the expected predominance of principal cells in the caput epithelium because together they contribute 1,369 cells compared with 2,192 principal cells in the combined analysis (Table S1). In earlier work in the rat epididymis (Trasler et al, 1988), principal cells were thought to account for about 75% of the total number. Whether our observation of a smaller contribution of principal cells in the human epididymis is due to species differences, or merely reflects cell recovery bias or the higher resolution of the scRNA-seq protocol, will become clear with more datasets from other species. It is of interest that these two cell types cluster closely together on the UMAP plot (Fig 2A), suggesting a more similar transcriptional signature in comparison with other cell types in the epithelium. Other

secreted proteins encoded by cluster 1 cells are EDDM3A and EDDM3B, which are poorly characterized, but also thought to play a key role in sperm maturation (Kirchhoff et al, 1994). Of note, deletion of these genes has been implicated in some cases of idiopathic azoospermia (Damyanova et al, 2013; Dong et al, 2015). Also notable in cluster 1 cells are DEGs from another gene family involved in the innate immune response, WFDC8, WFDC9, WFDC2, and WFDC11 which all map to chr20q13.12 and some of which are also androgen dependent.

Another aspect of the biology of the epididymis which our single cell atlas may help resolve is the identity of stem cells in the genital duct epithelium. Here, we can learn from the stem cells defined in other epithelia, based upon their capacity to generate organoids from a single cell. In the intestine, Lrg5+ cells in the crypt alone can generate functional organoids, which reproduce many key intestinal functions (Clevers, 2016). In the airway, TP63 positive basal cells are thought to be the stem cell population (Zuo et al, 2015) although there remains some controversy about this. Epididymis organoids were generated from single cells in several species (Mandon et al, 2015; Pinel et al, 2019; Leir et al, 2020), confirming the existence of stem cells within the epithelial cell population. These organoids are spherical structures, with a layer of basal cells on the outside and additional epithelial cells on the inside facing a luminal space. The scRNA-seq data presented here shows TP63 is a highly significant DEG among the basal cells in cluster 3, with a subpopulation having very high expression levels. This observation was confirmed by immunofluorescence, where p63 was abundant in a subset of keratin 5–expressing basal cells. With the exception of a few individual cells with TP63 transcripts in clusters 1, 2, and 4, this marker is not evident in other cell population in the epididymis suggesting the epididymis stem cells reside in the basal cell compartment.

Finally, it is highly relevant in terms of our understanding of CF pathology and how this disease causes male infertility to revisit the cells that express CFTR in the epididymis. CF impairs the function of many epithelia in different organs. It is notable that many surface epithelial cells in the digestive tract, both in the pancreatic ducts and the intestinal crypts, express abundant CFTR. However, in the lung CFTR-transcripts are at very low levels in most cells in surface epithelium with the exception of the CFTR "high" cells now identified as "pulmonary ionocytes" (Montoro et al, 2018; Plasschaert et al, 2018). Earlier elegant immunofluorescence data suggested that ciliated cells in the surface epithelium of the airways were the main site of CFTR protein (Kreda et al, 2005), so the scRNA-seq results were not consistent. We previously showed abundant CFTR mRNA and protein in cultured human caput epididymis epithelial (HEE) cells (Harris & Coleman, 1989; Leir et al, 2015; Browne et al, 2016b), although not in corpus and cauda HEE cells. We also found much lower levels of CFTR mRNA in caput tissue (Browne et al, 2016b), suggesting that the cells expressing CFTR were low abundance in the intact tissue or that in vivo CFTR expression was indeed lower in the same cell types. As for observations in the airway, our scRNA-seq data are similarly not consistent with recent immunofluorescence data on the human epididymis, where CFTR protein was shown to be high in principal cells along the duct (Sharma & Hanukoglu, 2019). It is possible that lack of specificity of the anti-CFTR antibodies may underlie these differences. Here, we show that in the caput epididymis, most principal cells, which are the most abundant surface epithelial cell type, express little or no

CFTR mRNA. In contrast, clear cells (cluster 6), which express high levels of genes encoding the V-ATPase hydrogen pump along with ENaC, the epithelial sodium channel, are also the primary location of abundant CFTR. These observations are consistent with the epididymis clear cell having an equivalent role to the "pulmonary ionocyte" in the airway epithelium. In the caput epididymis, we suggest that the primary role of CFTR in these sites is rapid co-ordination of the luminal environment that is required for normal sperm maturation. In contrast, in the EDs, where a population of surface epithelial cells, probably principal cell-like, express abundant CFTR, the protein may be involved in the main functions of the ED in water reabsorption (Sullivan et al, 2019). Accordingly, the high levels of CFTR in cultured HEE cells could either reflect the adaptation of principal cells to the submerged culture environment, or be in part due to minor contamination with ED cells. Either way, the dual role of CFTR in the proximal male genital ducts provide at least two mechanisms, whereby loss of CFTR could lead to epididymis abnormalities, absence of the vas deferens, and associated infertility in CF males.

## Materials and Methods

### Tissue

Human epididymis tissue was obtained with Institutional Review Board (IRB) approval from consented patients undergoing inguinal radical orchiectomy for a clinical diagnosis of testicular cancer. These are normal epididymis tissues and not pathological specimens because none of the epididymides have extension of the testicular cancer and donors are not receiving hormone therapy. Data included in this article were derived from seven donors aged 24 (2), 31, 32, 38, 47, and 57.

### Single cell isolation

Connective tissue and fat were removed from epididymis tissue within ~2 h of surgery and the tissue separated into EDs, caput, and corpus according to anatomical features (Fig 1A). Of note there was substantial divergence in the morphology of tissue from different donors making an exact delineation of regions difficult. Caput tissue was cut into 1–2 mm pieces and digested with collagenase (2 mg/ml collagenase + 150 µg/ml DNAse I, both from Worthington Biochemical Corp.) at 37°C for 2 h with constant shaking. Digesting tissue was then agitated by pipetting, allowed to settle for 4 min and the supernatant collected and stored on ice as cell suspension 1. Fresh collagenase solution was then added to the tissue, which was digested for a further 2 h with shaking at 37°C, with subsequent agitation and settling as before. The supernatant (cell suspension 2) was then pooled with cell suspension 1 and tissue debris removed by passage through a 100-µm cell strainer (Pluristrainer). After centrifugation (300$g$) of the cells, they were washed once in PBS and then digested with Accutase (STEMCELL Technologies) for 20–30 min. At this point, the cells were resuspended by gentle pipetting and examined by phase-contrast microscopy and if cell clumps were still present the Accutase digestion was repeated by

adding fresh Accutase after centrifugation. Once single-cell digestion was complete the cell pellet was lysed with ammonium chloride solution (310 mM $NH_4Cl$, 23.8 mM $NaHCO_3$, and 0.2 mM EDTA) for 3 min followed by adding 2.5× volume of PBS + 2% FBS and centrifuged at $300g$ for 5 min to remove red blood cells. Epithelial cells collected were washed twice in Hank's Balanced Salt Solution + 2% FBS with centrifugation at ($200g$), then resuspended in Hank's Balanced Salt Solution and passed through Flowmi cell strainers (40 $\mu$m; Bel-Art) to prepare single cells. Cells were counted and then used for scRNA-seq.

### scRNA-seq and analysis

A total of 2,500–3,000 cells from each of three donor samples were used for scRNA-seq using the 10× Genomics Chromium Single Cell 3′ Reagent Kit (v2). After Tapestation quality control, libraries were sequenced on a NovaSeq 600 sequencer (~300 million reads). Library reads were aligned to the hg19 genome package v1.0 using Cell Ranger 3.1.0, then the cell–UMI matrix was exported into Seurat V3. The matrix was subsequently filtered with min.cells = 3, min.features = 200. The mitochondrial reads ratio median was 0.0212, and the third quantile was 0.0324, well below the recommended 0.05. The three biological replicates were merged and 5,299 single-cell transcriptomes obtained post filtering, and corrected for batch effect using Seurat v3 integration with 30 dimensions and 20,000 anchor features. Cell neighbors were then found using 10 dimensions and unsupervised clustering at a resolution of 0.06. Thirty PCA dimensions were reduced using UMAP. The Seurat v3 package was also used to perform differential gene expression analysis and generate plots, including violin plots, ridge plots, and feature plots. Both FindAllMarkers and heat map generation used myAUC statistics methods. Cell Ranger 3.1.0 output data were also analyzed and visualized in the Loupe Cell Browser V3.1.1. Sequence data are deposited at GEO:GSE148963.

### Immunofluorescence microscopy

Paraformaldehyde-fixed and paraffin-embedded epididymis tissues were cut into 5-$\mu$m sections. After deparaffinization and rehydration, antigen retrieval was performed in sodium citrate buffer (10 mM sodium citrate and 0.05% Tween 20, pH 6.0) at 98°C water bath for 45 min. The sections were then post-fixed in 4% paraformaldehyde (in PBS) for 15 min, permeabilized with 0.05% saponin for 10 min, and blocked with 1% BSA before staining. The sections were then incubated with primary antibody/antibodies at 4°C overnight. Primary antibodies used were AR (sc-816; Santa Cruz Biotechnology), ESR1 (sc-8002), ATP6V1B1/B2 (ab200839; Abcam), CFTR (ab2784), KRT5 (ab52635), TP63 (ab735), VIL1 (ab130751), CST11 (HPA053399; Sigma-Aldrich), CRISP1 (HPA028445), DEFB118 (HPA042634), SPAG11B (HPA023842), SPINK13 (HPA036456), and WFDC8 (HPA071119). After three washes with PBS (with 0.05% Tween 20), the sections were incubated with secondary antibody for 1 h at room temperature. Secondary antibodies were Alexa Fluor 488–conjugated antirabbit IgG and Rhodamine Red-X–conjugated antimouse IgG (both from Jackson ImmunoResearch). After washing, the samples were counterstained with DAPI for nuclear identification, mounted with prolong antifade mountant (Invitrogen), and analyzed using a Leica DMR 6000 microscope. Images were taken with 20× (numerical aperture = 0.5) or 40× (numerical aperture = 1.25–0.75) objective lenses at room temperature. A Q-Imaging Retiga

Xi FireWire high-speed, 12-bit cooled CCD camera with an IR blocking filter was used to take the images with Volocity (ver. 6.3; PerkinElmer) as acquisition software to generate 8-bit tiff image files.

### Supplemental material

The supplemental material includes six figures showing data pertinent to the results and discussion section; also two tables with details of scRNA cell counts (1) and DEG lists for each cell cluster identified in the scRNA-seq analysis by the Seurat pipeline (2).

### Ethical approval

Procedures were performed according to the Case Western Reserve University Research Committee IRB protocol #2017-2099. Informed consent was obtained from all tissue donors.

## Supplementary Information

## Acknowledgements

The authors thank Dr Katrina Diener and colleagues at the University of Colorado Anschutz Medical Campus Genomics Core for 10× sequencing; Dr Brian Cobb for advice; the Case Western Reserve University School of Medicine (CWRU SOM) Light Microscopy Core Facility (NIH Grant S10RR021228), the Cleveland Clinic Central Biorepository, and PLMI for providing epididymis tissues; and the Department of Genetics and Genome Sciences, CWRU, and the Case Comprehensive Cancer Center for infrastructure support. Funding: This work was supported by the Cystic Fibrosis Foundation (Leir17G0, Harris 18P0) and the National Institutes of Health R01 HD068901 and R01 HL094585 (both to A Harris).

### Author Contributions

S-H Leir: conceptualization, resources, data curation, formal analysis, funding acquisition, validation, investigation, visualization, methodology, project administration, and writing—original draft, review, and editing.
S Yin: data curation, software, formal analysis, validation, investigation, methodology, and writing—review and editing.
JL Kerschner: data curation, investigation, methodology, and writing—review and editing.
W Cosme: data curation and formal analysis.
A Harris: conceptualization, data curation, formal analysis, supervision, funding acquisition, validation, investigation, visualization, methodology, project administration, and writing—original draft, review, and editing.

### Conflict of Interest Statement

The authors declare that they have no conflict of interest.

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
