## [Reviewer comments · Life Science Alliance]

An atlas of human proximal epididymis reveals cell-specific functions and distinct roles for CFTR

Shih-Hsing Leir, Shiyi Yin, Jenny Kerschner, Wilmel Cosme and Ann Harris
DOI: 10.26508/lsa.202000744

Corresponding author(s): Prof. Ann Harris (Case Western Reserve University)

Review timeline:

Submission Date:	2020-04-17
Editorial Decision:	2020-06-04
Revision Received:	2020-07-24
Editorial Decision:	2020-08-04
Revision Received:	2020-08-06
Accepted:	2020-08-13

Scientific Editor: Shachi Bhatt

Transaction Report:

No Peer Review Process File is available with this article, as the authors have chosen not to make the review process public in this case.

Re: Life Science Alliance manuscript #LSA-2020-00744-T

Prof. Ann Harris
Case Western Reserve University
Genetics and Genome Sciences
10900 Euclid Ave
Cleveland, OH 44106

Dear Dr. Harris,

Thank you for submitting your manuscript entitled "An atlas of human proximal epididymis reveals cell-specific functions and distinct roles for CFTR" to Life Science Alliance. The manuscript was assessed by expert reviewers, whose comments are appended to this letter.

We would like to invite a revision of your manuscript. The dataset could be useful for the community, however the points raised by the reviewers, especially reviewer #3 have to be addressed. Also the figures should be consolidated and the data should be made available in an easy to use form since the main value of the work will be the atlas data. In that sense, it will be crucial to provide easily accessible raw and processed data. With regards to comparing the atlas to the murine system, which both reviewers requested, I think a superficial comparison would be good enough (i.e. comparing marker genes of the mouse in the human clusters) and would not expect a true integrative comparison of the data with the mouse data since this is still a non-trivial task.

Thank you for this interesting contribution to Life Science Alliance. We are looking forward to receiving your revised manuscript.

Sincerely,

Reilly Lorenz

Editorial Office Life Science Alliance
Meyershofstr. 1
69117 Heidelberg, Germany
t +49 6221 8891 414
e contact@life-science-alliance.org
www.life-science-alliance.org

B. MANUSCRIPT ORGANIZATION AND FORMATTING:

RE: Life Science Alliance Manuscript #LSA-2020-00744-TR

Prof. Ann Harris
Case Western Reserve University
Genetics and Genome Sciences
10900 Euclid Ave
Cleveland, OH 44106

Dear Dr. Harris,

Thank you for submitting your revised manuscript entitled "An atlas of human proximal epididymis reveals cell-specific functions and distinct roles for CFTR". Your manuscript was re-reviewed by the original referees, and their reports are attached below. We would be happy to publish your paper in Life Science Alliance pending final revisions necessary to meet our formatting guidelines.

- please add a conflict of interest statement to the main manuscript text
- please upload your supplementary figures as single files
- please add a callout in the main manuscript text for figures: Fig. 4J, Fig. 5B, 5D, 5F, Fig. 6B, Fig. 7B, 7D, Suppl. Fig. S1A-D, Suppl. Fig. 5C
- please make sure that the figure panels are in alphabetical order in the figure legends
- to make the panels in the figure legends more clear, please de-emphasize the abbreviations (eg '...the efferent ducts (ED)...' to 'the efferent ducts {ED}..' or put all abbreviations at the end of the legend
- please add the supplementary figure legends to the main manuscript text
- please upload tables as editable doc or excel files
- please use the [10 author names, et al.] format in your references (i.e. limit the author names to the first 10)
- please add scale bars to Figure S1
- in your figures, please position the panel indicators ('A', 'B'...) at the upper left of each figure and increase the font size

A. FINAL FILES:

B. MANUSCRIPT ORGANIZATION AND FORMATTING:

Sincerely,

Reilly Lorenz
Editorial Office Life Science Alliance
Meyerhofstr. 1
69117 Heidelberg, Germany
t +49 6221 8891 414
e contact@life-science-alliance.org
www.life-science-alliance.org

3rd Editorial Decision

13 August 2020

RE: Life Science Alliance Manuscript #LSA-2020-00744-TRR

Prof. Ann Harris
Case Western Reserve University
Genetics and Genome Sciences
10900 Euclid Ave
Cleveland, OH 44106

Dear Dr. Harris,

Thank you for submitting your Research Article entitled "An atlas of human proximal epididymis reveals cell-specific functions and distinct roles for CFTR". It is a pleasure to let you know that your manuscript is now accepted for publication in Life Science Alliance. Congratulations on this interesting work.

DISTRIBUTION OF MATERIALS:

Again, congratulations on a very nice paper. I hope you found the review process to be constructive and are pleased with how the manuscript was handled editorially. We look forward to future exciting submissions from your lab.

Sincerely,
Shachi

Shachi Bhatt
Executive Editor, Life Science Alliance
www.life-science-alliance.org